# The stress reduction potential of Bhagavad Gita and Yoga for healthcare workers during the COVID-19 pandemic: A randomized controlled trial

**Nishant Das[1‡], Laalithya Konduru[2,3,4‡]\*, Simranjeet Singh Dahia[3,5], Santwana Sagnika[6], Gargi Kothari-Speakman[7], Ajit Kumar Behura[1]**

1 Department of Humanities and Social Sciences, Indian Institute of Technology (ISM), Dhanbad, Jharkhand, India, 2 Department of Community Medicine, Sri Jagannath Healthcare and Research Center, Dhanbad, Jharkhand, India, 3 College of Medicine and Public Health, Flinders University, Bedford Park, South Australia, Australia, 4 Harvard T.H. Chan School of Public Health, Harvard Univresity, Boston, Massachusetts, United States of America, 5 Mineral Resources, Commonwealth Scientific and Industrial Research Organisation, Waite Campus, Urrbrae, South Australia, Australia, 6 School of Computer Engineering, Kalinga Institute of Industrial Technology, Bhubaneswar, Odisha, India, 7 Savitri Ghantasala Center for Health Equity, Samanjasa Foundation, Chennai, Tamil Nadu, India

‡ These authors must be considered joint first authors on this work.
* laalithya@gmail.com

## Abstract

### Background

Healthcare workers (HCWs) experienced elevated psychological distress during the COVID-19 pandemic. Culturally embedded, non-stigmatizing approaches may improve uptake of mental health support in this population. We aimed to evaluate the impact of Yoga, Bhagavad Gita learning, and their combination (Yoga+Gita) on psychological distress among HCWs.

### Methods

We conducted a prospective, single-center, parallel-group randomized controlled trial at a secondary hospital in India. Eligible HCWs (>18 years, without pre-existing diagnosed mental health conditions) were randomized in a 1:1:1:1 ratio to: Yoga (daily 60-min sessions for one week), Gita (daily 60-min teaching of Bhagavad Gita Chapter 2 for one week), Yoga+Gita (30 min Yoga + 30 min Gita daily for one week), or Control (usual care). Psychological distress was assessed using the Generalized Anxiety Disorder-7 (GAD7) scale at baseline (B-GAD7), immediately post-intervention (IPI-GAD7), and 45 days post-intervention (DPI-GAD7). Participants were aware of group assignment, but data analysts were blinded. Group comparisons used Friedman and Kruskal–Wallis tests with Bonferroni-adjusted post-hoc analyses. We hypothesized that (1) all three interventions would reduce GAD-7 scores compared with control, (2) Gita would produce more durable effects than Yoga

**Data availability statement:** Data collected for the study, including individual participant data and a data dictionary defining each field in the set, has been deposited in Figshare, with the doi: https://doi.org/10.6084/m9.figshare.27285234. The Figshare dataset is publicly accessible without restriction. The study protocol and statistical analysis plan have been registered in the OSF Registry, with the doi: https://doi.org/10.17605/OSF.IO/PXG93.

**Funding:** The author(s) received no specific funding for this work.

**Competing interests:** The authors have declared that no competing interests exist.

alone, and (3) the combined Yoga+Gita intervention would yield the greatest overall reduction in psychological distress.

## Findings

Sixty-eight participants were enrolled (17 per group), with no losses to follow-up. B-GAD7 did not differ across groups (p = 0.908). Compared with Control, Yoga+Gita reduced anxiety immediately (p = 0.009), and Gita reduced anxiety at 45 days (p = 0.047). A clinically meaningful (≥4-point) GAD7 reduction at 45 days occurred in 17.6% (Yoga), 47.1% (Gita), 41.2% (Yoga+Gita), and 11.8% (Control) of the participants. Exploratory analyses suggested larger benefits among women and among those who continued practice.

## Interpretation

Yoga and Gita-learning, particularly in combination, were associated with reductions in psychological distress among HCWs.

## Trial registration

Prospectively registered with the CTRI (CTRI/2021/10/037365).

---

## 1. Introduction

To effectively combat public health challenges, it is important for the frontline workers to remain healthy. According to the World Health Organization (WHO), the psychological wellbeing of healthcare workers (HCWs) is as important as their physical health [1]. Globally, HCWs reported increased mental health problems during the coronavirus disease (COVID-19) pandemic [2].

The WHO advised the HCWs to seek social support for maintaining mental wellbeing [1]. The managers of health facilities were instructed to facilitate access to mental health support services for staff [1]. Thus, the need for mental health support for HCWs has been recognized; however, no specific strategy has been devised by the medical fraternity to address the HCWs' mental health concerns [3]. Patel et al. recommend engaging in meditation and yoga to manage stress in HCWs, and approaching therapists if required [4]. Li et al. discuss how online counselling is being adopted to provide mental health support to HCWs and patients in China [5]. Another study summarizes some of the practices adopted by different countries to address the psychological impact of COVID-19 on HCWs, including measures like peer support programs, online courses on common psychological problems, facilitating counselling at staff rest areas, and a dedicated helpline to provide advice and psychological support to the HCWs [6]. Ho et al. suggest shorter and rotating shifts and regular breaks for frontline HCWs [7]. Kar et al. recommend healthy lifestyle measures and pursuing hobbies [3].

However, despite psychological distress, most HCWs decline seeking help [2,8,9], rendering the mental health support services unproductive. Research suggests that

HCWs do not access mental health services due to unrealistic expectations of themselves, denial and minimization of symptoms [2,9,10]. Therefore, mere creation of mental health support systems may be of limited help, and facilitation of their access by the managers may be difficult as HCWs do not seek help.

Culturally oriented health services, including mental health services, are known to improve quality of care [10]. To make mental health services culturally appropriate, socio-constructive approaches that emphasize the cultural context of an individual's experience and reality have been adopted [11]. Despite recognizing the need for culturally sensitive mental health services, there is no specific guidance from the international medical fraternity on culturally tailoring mental health services to HCWs. This is especially problematic for a country like India, where the cultural context of the patients and the HCWs differs considerably from the West. The bulk of the research on mental health has been carried out in Western countries, and it cannot be said with certainty how successfully the findings can applied to other cultures [12]. Therefore, evidence-based culturally-sensitive programs must be designed [12].

During the peak of the first wave of the COVID-19 pandemic in India, we instituted mental support measures, such as access to a confidential counselling and psychology service and instituting a peer support group, for HCWs at our hospital. However, the uptake of these services was low—out of 150 HCWs working at our hospital, only 3 HCWs accessed the counselling and psychology service and 5 HCWs participated in the peer support group between 16, September 2020 and 17, March 2021—and it was unviable for our hospital to continue providing these services. Subsequent research conducted at our hospital revealed that there was significant stigma attached to seeking mental health support and that most HCWs turned to spirituality as a coping mechanism [2]. Thus, recognizing the value of culturally-oriented services, which may be rooted in spirituality in order to enhance uptake, herein, we attempt to validate the postulations of a rather oriental and emic approach by Das & Behura based on imparting the teachings of the Bhagavad Gita—a Hindu scripture—to the HCWs alongside conventional measures [13]. The Bhagavad Gita is a popular source of knowledge and wisdom. The narration of the Gita, set in the middle of a battlefield, provides direction and lifts the spirits of a warrior (Arjuna), who is waging an inner battle alongside the external one. Das & Behura hypothesized that the Bhagavad Gita can help the HCWs, just like it helped Arjuna, despite the cause of distress of the HCWs having no moral bearings, unlike the one that plagued Arjuna [13]. The specific hypotheses evaluated in this study are as follows:

(1) Practicing yoga, learning the Bhagavad Gita, or combining the two will lead to significant reductions in psychological distress compared to a control group; (2) learning the Bhagavad Gita will have more lasting effects on stress reduction than Yoga alone; and (3) Yoga and Bhagavad Gita will act synergistically to provide both immediate and long-term stress reductions.

## 2. Methods

This randomized controlled trial is reported in accordance with the CONSORT 2010 checklist, which is provided in the Supporting Information (See S1 File).

### 2.1. Study setting

The study was conducted at Sri Jagannath Healthcare And Research Center (SJHRC), a secondary hospital in Dhanbad, Jharkhand, India, between the second and third waves of the COVID-19 pandemic in India. Jharkhand has a doctor–patient ratio of 1:8200. Participant recruitment commenced on 11th October, 2021 and was completed on 23rd October, 2021.

### 2.2. Ethical considerations, participant recruitment, and study design

The study and all the procedures were approved by the SJHRC – Independent Ethics Committee (approval number: SJHRC – NI/21/OCT/07). The trial was prospectively registered on the Clinical Trials Registry of India (CTRI/2021/10/037365). A call for participants was posted on the internal mail system and notice board of SJHRC, along

with all relevant information on the study. The study remained open for enrollment for 2 weeks. When a HCW showed interest in participating in the study, they were invited to an interview with the research team where the research team again explained the objectives and methods of the study and the HCWs clarified any questions they had about the study; then the HCWs were assessed against the inclusion and exclusion criteria, and written informed consent was obtained from those deemed eligible for the study. The inclusion criteria were age > 18 years, current employment at SJHRC as an HCW, and availability for the duration of the interventions and follow-up. The exclusion criterion was the presence of any pre-existing diagnosed mental health condition; this was to ensure that the study measured the impact of the interventions on psychological distress arising from work-related stress during the pandemic and not underlying disorders.

Upon returning the signed consent form to the research office, the participants were randomized in a 1:1:1:1 ratio using a computer-generated random sequence. The sequence, containing participant identification numbers (PINs) and corresponding group assignments, was generated in advance by a researcher not involved in recruitment and placed into sequentially numbered, opaque, sealed envelopes that were handled only by research staff uninvolved in enrollment. The sealed envelopes were placed in a container prior to enrolment. After written informed consent was obtained, each participant selected one envelope from the container, which was then opened to reveal their group assignment. This procedure ensured both random sequence generation and allocation concealment at the time of enrollment. All data was collected anonymously using the PIN as the only identifier. The consent forms and a document linking the names of the participants to their PIN was kept under lock and key at the research office of SJHRC; the research team of this study did not have access to this document.

A prospective, randomized controlled repeated measures design was employed in this study. The trial protocol was registered on the OSF registry (https://doi.org/10.17605/OSF.IO/PXG93) and is provided in the Supporting Information (see S2 File). The novel intervention (imparting the teachings of the Bhagavad Gita alongside practicing yoga) was tested against a standard intervention (practicing yoga). We also tested the efficacy of imparting the teachings of the Bhagavad Gita as a standalone intervention. Thus, there were four study arms: Control—the participants received no intervention, Yoga—the participants attended an hour-long yoga session every day for one week, Gita—the participants attended an hour-long Bhagavad Gita session every day for one week, Yoga+Gita—the participants attended an hour-long session where they were made to practice yoga for the first 30 mins, and the Bhagavad Gita was taught over the remaining 30 mins.

Verses from the second chapter of the Bhagavad Gita were taught and their meaning was explained by trained teachers of International Society for Krishna Consciousness (ISKCON), Dhanbad, using the text Bhagavad-Gita As It Is as part of the intervention to participants in the Gita and Yoga+Gita groups. Chapter 2 was selected because it synthesizes core teachings on detachment, duty, and mind-control, providing a practical cognitive framework for stress regulation and resilience. Printed handouts of the verses were provided, the same instructor delivered all sessions, and participants were encouraged to review the verses during the 45-day follow-up period. Each 60-minute Yoga session (Yoga group) followed a fixed, standardized format consisting of approximately 30 minutes of asana postures, 15 minutes of pranayama, and 15 minutes of guided meditation. The 30-minute Yoga sessions in the Yoga+Gita arm followed the same sequence, with each component proportionally reduced in duration. Participants in the Yoga and Yoga+Gita groups were taught and made to practice the following poses and breathing exercises by trained teachers of Ayush and Skill Development Research Institute, Dhanbad: Bhadrasana (Gracious Pose), Mandukasana (Frog Pose), Shawasana (Corpse pose), Paschimottanasana (Seated Forward Bend Pose), Naadishodhana Pranayama (Alternate nostril breathing), Bhramari Pranayama (Bee breath), Sheetkari Pranayama (Hissing Breath), Sheetali Pranayama (Cooling Breath), and Dhyana (Meditation). The same instructors conducted all Yoga sessions throughout the intervention period.

### 2.3. Randomization and masking

Participants were randomized as described in Section 2.2, using a computer-generated allocation sequence with concealment via sequentially numbered, opaque, sealed envelopes. The research teams were separated into three groups: Team

1 handled recruitment and enrollment; Team 2 coordinated the interventions and collected outcome data; and Team 3 conducted data analysis. The three teams did not have contact with each other. Due to the behavioral nature of the interventions, participants and instructors were aware of group assignments. However, personnel conducting data analysis (Team 3) were blinded to treatment allocation.

## 2.4. Measures

The Generalised Anxiety Disorder-7 (GAD7) was used to measure psychological distress as it is sensitive to change over a short period [14]. It has also been used by previous studies to measure psychological outcomes in HCWs during the COVID-19 pandemic [15]. One week after the last participant had enrolled in the study, the GAD7 was administered to all participants; this was the baseline assessment (B-GAD7). Along with the B-GAD7, demographic data (age, gender, and education) was also collected from the participants. The pertinent interventions were started on the next day for the three intervention groups. On the eighth (1 day post intervention) and the fifty-second (45 days post-intervention) days after the start of the interventions, the GAD7 was again administered to all groups, including the Control group; these were called the immediate post-intervention GAD7 (IPI-GAD7) and the delayed post-intervention GAD7 (DPI-GAD7), respectively. While collecting the DPI-GAD7 data, the participants in the intervention groups were also asked a single yes/no question: if they had continued the practices taught during the intervention; no data were collected on frequency, duration, or fidelity of continued practice.

## 2.5. Statistical analysis

PASS 2020, v20·0·6 (NCSS LLC, UT, USA) was used to calculate the sample size based on a two-sided two-sample t-test of mean change in GAD7 scores for an α of 5%, 1-β of 80%, assuming a minimum clinically important difference of 4 points for the GAD7 [16], a standard deviation (SD) of 3·78 based on a previous study of one of the interventions of interest [17], and a dropout rate of 15%; this model was used for power only. This calculation targeted a two-group intervention-versus-control comparison and did not incorporate multiplicity adjustment for all pairwise comparisons among the four arms. Data were analyzed using Python 3·9. The Shapiro–Wilk test was used to test the normality of the data for age and for GAD7 within each group at each time point (baseline, IPI, DPI). Age and GAD7 scores were not normally distributed. Between groups at baseline, age was compared with Kruskal-Wallis, while gender and education, being categorical, were compared using $\chi^2$ tests. To compare GAD7 scores within the groups over time and across the groups, the Friedman and Kruskal–Wallis tests were utilized, respectively. Following a significant Friedman test result, pairwise comparisons were conducted using the Conover-Friedman post-hoc test. For these Conover-Friedman post-hoc comparisons, p-values were adjusted using a Bonferroni correction (α_adj = 0.05/k, where k is the number of pairwise comparisons within each test family). If the Kruskal–Wallis test indicated significant differences, Dunn's pairwise tests with Bonferroni-adjusted two-sided p-values were used for pairwise comparisons. To measure the effect size of the differences in GAD7 values between groups, $\eta^2$ was reported (for Kruskal-Wallis). The Kendall's W was calculated for Friedman to measure the effect size of the differences in GAD7 values within groups over time. We summarized the proportion achieving the MCID (≥4-point reduction from Baseline to DPI). For exploratory estimation of clinical effect, odds ratios and 95% confidence intervals were calculated using exact methods for 2×2 contingency tables (conditional maximum likelihood estimation), given the small cell counts, comparing MCID responder status in each intervention arm versus control. Percent change was computed per participant as 100×(DPI–Baseline)/Baseline (participants with Baseline = 0 excluded), summarized as median [IQR]. Additionally, to determine the effect of continued practice of the intervention, which is pivotal for understanding the sustained impact of the interventions on psychological wellbeing, we divided each group into subgroups based on whether the participants continued with the practices taught during the intervention, and employed the Mann–Whitney U test (exploratory) to compare the DPI-GAD7 scores among the subgroups. Exploratory subgroup analyses were also conducted based on gender and educational attainment, and the Mann–Whitney U test was utilized to

compare the B-GAD7, IPI-GAD7, and DPI-GAD7 scores between the subgroups. Given limited power, all continued-practice and subgroup analyses were exploratory and not adjusted for multiple comparisons; their results should therefore be interpreted as hypothesis-generating, whereas primary post-hoc comparisons (Dunn's and Conover–Friedman tests) were Bonferroni-adjusted and are reported with corresponding effect sizes ($\eta^2$ and Kendall's W). A p-value <0·05 was deemed significant for all statistical tests. Between-group tests used all available observations at each timepoint; Friedman/Conover–Friedman used complete cases with all three timepoints within arm; MCID analysis required both Baseline and DPI-GAD7 scores.

## 3. Results

### 3.1. Timeline of interventions

The B-GAD7 was administered on 1st November, 2021 and the interventions ran between 2nd November, 2021 and 8th November, 2021. The IPI-GAD7 and DPI-GAD7 were administered on 9th November, 2021 and 23rd December, 2021, respectively. There were 17,948 COVID-19-related deaths in India between 1st November, 2021 and 23rd December, 2021 [18].

### 3.2. Study participants

The sample size calculation revealed that 68 participants, with 17 participants per group, were required to yield an α of 5% and 1-β of 80%. There were no losses or exclusions after randomization. All participants were included in the analysis and the analysis was by original assigned groups. All participants had at least a Bachelor's degree. There were no statistically significant differences between the groups at baseline in age (Kruskal–Wallis p=0·938), educational attainment ($\chi^2$ p=0·891), or gender ($\chi^2$ p=0·706). The demographic characteristics of the participants is presented in Table 1. The CONSORT flowchart of the study is shown in Fig 1.

### 3.3. Changes in GAD7 values

Table 2 shows the median (interquartile range) of the GAD7 scores over time in each group. The results illustrate a marked reduction in anxiety levels for participants in the Gita and Yoga+Gita groups.

   3.3.1. **Across groups comparison.** The Kruskal–Wallis test showed no significant differences in the B-GAD7 values across the groups (p=0·908). However, it showed significant differences in the IPI-GAD7 (p=0·011; $\eta^2$=0·166; Dunn's test significant between the Yoga+Gita and Control groups, p=0·009) and DPI-GAD7 (p=0·025; $\eta^2$=0·140; Dunn's test significant between the Gita and Control groups, p=0·047). Following Cohen's convention, $\eta^2$ values ≥0·14 were interpreted as large effects, indicating a meaningful between-group separation at both post-intervention timepoints. The detailed results of the Dunn's test are shown in Table 3. Compared with Control, the odds of achieving MCID were higher in the Gita group (OR=6.67, 95% CI 1.15–38.60) and in the Yoga+Gita group (OR=5.25, 95% CI 0.90–30.62), whereas

**Table 1. Participant demographic characteristics.**

| Group | N | Age (Median [IQR]); years | Gender, n (%) | | Education, n (%) | |
|---|---|---|---|---|---|---|
| | | | **Men** | **Women** | **Bachelors** | **Masters or higher** |
| **Yoga** | 17 | 29·000 [24·000, 49·000] | 11 (64·700%) | 6 (35·300%) | 9 (52·900%) | 8 (47·100%) |
| **Gita** | 17 | 30·000 [26·000, 52·000] | 11 (64·700%) | 6 (35·300%) | 10 (58·800%) | 7 (41·200%) |
| **Yoga+Gita** | 17 | 37·000 [30·000, 45·000] | 8 (47·100%) | 9 (52·900%) | 10 (58·800%) | 7 (41·200%) |
| **Control** | 17 | 36·000 [30·000, 46·000] | 10 (58·800%) | 7 (41·200%) | 8 (47·100%) | 9 (52·900%) |

*IQR, interquartile range.*

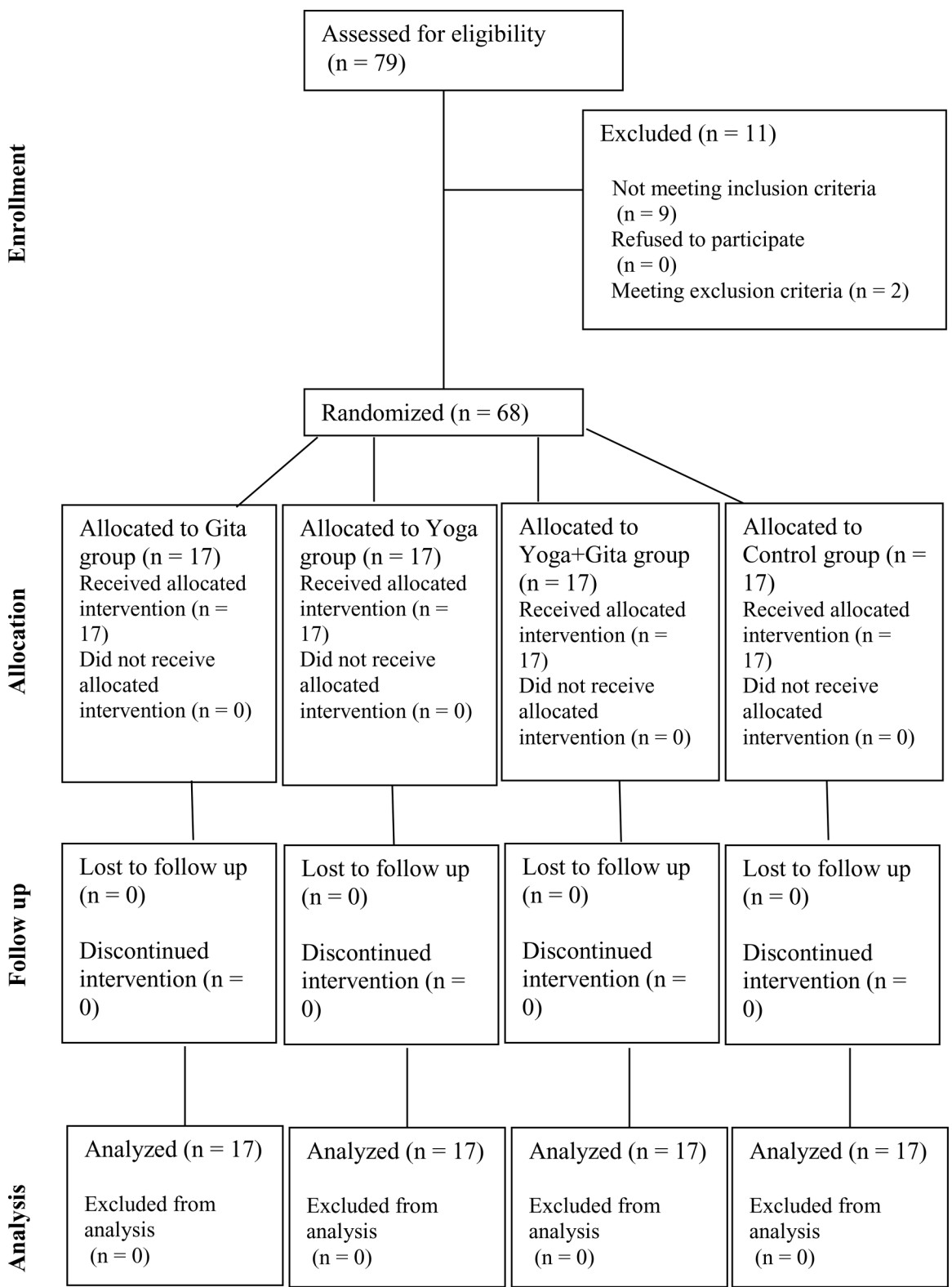

**Fig 1. CONSORT flow diagram showing the number of participants assessed for eligibility (n = 79), excluded (n = 11), randomized (n = 68), allocated to each arm, followed up, and analyzed (n = 17 per group).**

**Table 2. GAD7 values by group over time.**

| Group | B-GAD7 | IPI-GAD7 | DPI-GAD7 | %Change (Baseline to 45 Days) (Median %Change [IQR]) | MCID Responders | |
|---|---|---|---|---|---|---|
| | Median [IQR] | Median [IQR] | Median [IQR] | | n | % |
| **Yoga** | 6·000 [4·000, 9·000] | 5·000 [3·000, 7·000] | 5·000 [2·000, 6·000] | −30·000% [-50·000%, 0·000%] | 3 | 17·6% |
| **Gita** | 7·000 [5·000, 9·000] | 4·000 [3·000, 6·000] | 3·000 [2·000, 4·000] | −50·000% [-70·000%, -33·333%] | 8 | 47·1% |
| **Yoga+Gita** | 6·000 [4·000, 7·000] | 3·000 [2·000, 5·000] | 4·000 [1·000, 5·000] | −50·000% [-66·667%, -20·000%] | 7 | 41·2% |
| **Control** | 7·000 [4·000, 9·000] | 6·000 [4·000, 8·000] | 7·000 [4·000, 9·000] | −8·333% [-33·333%, 25·000%] | 2 | 11·8% |

*MCID, minimum clinically important difference (Responders are those with MCID >=4); GAD7, generalised anxiety disorder-7; B-GAD7, baseline GAD7; IPI-GAD7, immediate post-intervention GAD7; DPI-GAD7, delayed post-intervention GAD7; IQR, interquartile range.*

*MCID Responders are participants with ≥4-point reduction in GAD7 from Baseline to DPI (Day 52), denominator for calculating %=participants with both Baseline and DPI measurements. %Change is computed per participant as 100×(DPI−Baseline)/Baseline (participants with Baseline=0 excluded), then summarized as median [IQR].*

**Table 3. Results of the Dunn's test.**

| Timepoint | Comparison | Dunn's test p-value | Significant Difference |
|---|---|---|---|
| IPI-GAD7 | Yoga vs Gita | 1·000 | No |
| IPI-GAD7 | Yoga vs Yoga+Gita | 0·317 | No |
| IPI-GAD7 | Yoga vs Control | 1·000 | No |
| IPI-GAD7 | Gita vs Yoga+Gita | 1·000 | No |
| IPI-GAD7 | Gita vs Control | 0·153 | No |
| IPI-GAD7 | Yoga+Gita vs Control | 0·009 | Yes |
| DPI-GAD7 | Yoga vs Gita | 1·000 | No |
| DPI-GAD7 | Yoga vs Yoga+Gita | 1·000 | No |
| DPI-GAD7 | Yoga vs Control | 0·944 | No |
| DPI-GAD7 | Gita vs Yoga+Gita | 1·000 | No |
| DPI-GAD7 | Gita vs Control | 0·047 | Yes |
| DPI-GAD7 | Yoga+Gita vs Control | 0·057 | No |

*GAD7, generalised anxiety disorder-7; B-GAD7, baseline GAD7; IPI-GAD7, immediate post-intervention GAD7; DPI-GAD7, delayed post-intervention GAD7*

the Yoga group showed little difference (OR=1.61, 95% CI 0.23–11.09). Confidence intervals were wide due to the small sample size. The group medians over time is presented in Fig 2.

**3.3.2. Within group comparison.** The Friedman test showed no significant differences in the GAD7 values over time in the Control group (p=0·779). However, it showed significant differences in the Yoga (p=0·001; Kendall's W=0·422), Gita (p<0·001; Kendall's W=0·402), and Yoga+Gita (p<0·001; Kendall's W=0·410) groups. As per Cohen's convention, the Kendall's W values around 0·40 were interpreted as reflecting a moderate-to-large degree of concordance in GAD-7 change over time within these intervention arms. The Conover test showed significant differences between B-GAD7 and DPI-GAD7 in the Yoga (p=0·007) and Gita (p=0·003) groups, and between B-GAD7 and IPI-GAD7 (p=0·012) and B-GAD7 and DPI-GAD7 (p=0·011) in the Yoga+Gita group. The detailed results of the Conover test are shown in Table 4.

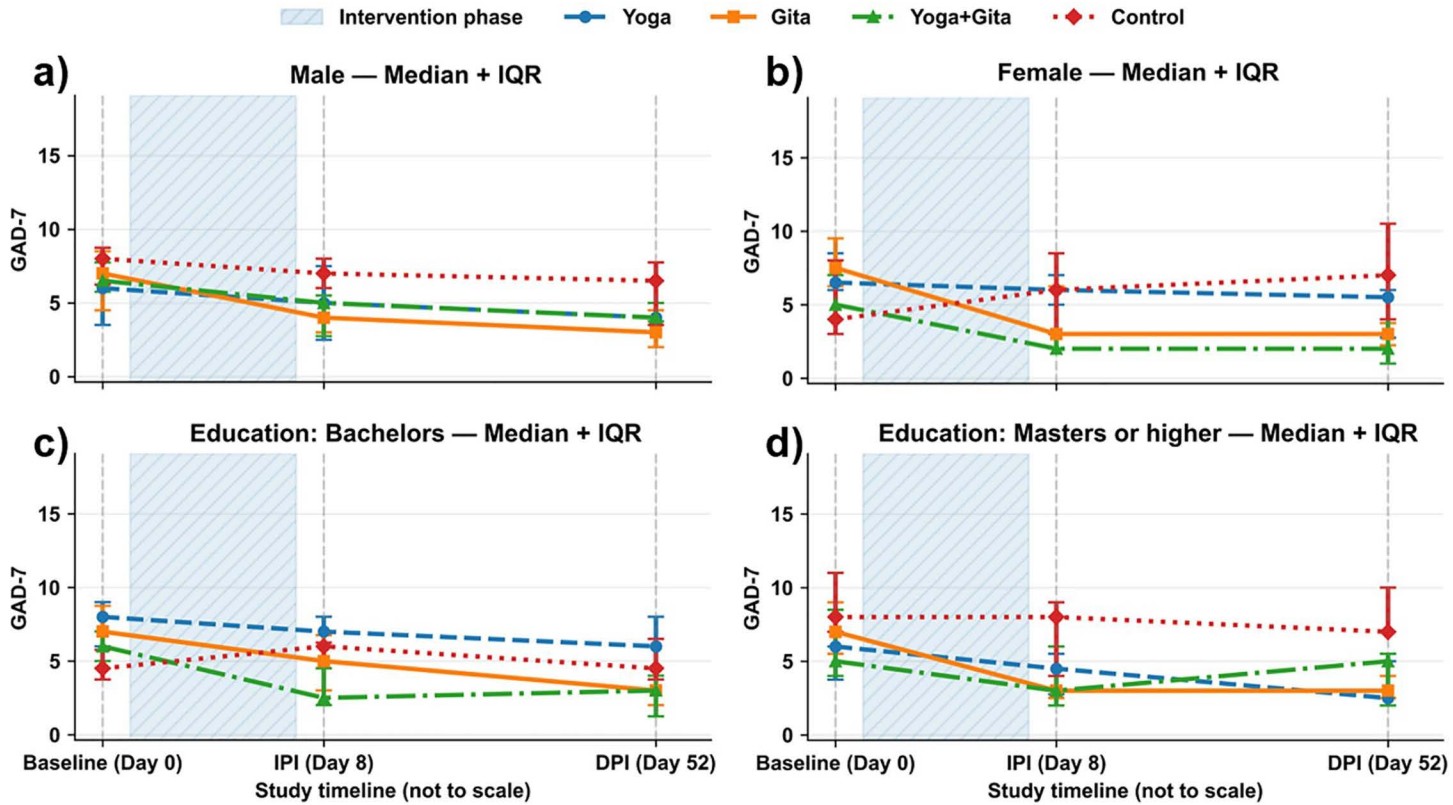

**Fig 2. Median (IQR) GAD7 scores over time by study group, stratified by gender and education level (exploratory).** Scores are shown at baseline (Day 0), immediately post-intervention (Day 8), and 45 days post-intervention (Day 52). The shaded region indicates the intervention period.

### 3.4. Subgroup analysis

All subgroup analyses presented below were exploratory and were not powered for confirmatory inference.

**3.4.1. By gender.** The data revealed that women experienced a more pronounced decrease in GAD-7 scores, especially within the Gita and Yoga+Gita groups. The median B-GAD-7, IPI-GAD7, and DPI-GAD7 scores along with the IQRs for both men and women across all groups are shown in Table 5.

Men in all three groups showed improvements in GAD7 scores over time (Friedman test: Yoga, p = 0·009; Gita, p = 0·003; Yoga+Gita, p = 0·002). However, the control group remained largely unchanged. Women demonstrated a more substantial reduction in GAD7 scores, particularly in the Gita (Friedman test, p = 0·006) and Yoga+Gita (Friedman test, p = 0·002) groups. However, the Yoga group did not show a significant difference (Friedman test, p = 0·108). Notably, the control group showed an increase in DPI-GAD7 scores in women, although not significant (Friedman test, p = 0·618), contrasting with the intervention groups' outcomes.

**3.4.2. By Education Level.** The median B-GAD-7, IPI-GAD7, and DPI-GAD7 scores for participants with a Bachelors degree and those with a Masters degree or higher, along with the IQRs, across all groups are shown in Table 6.

For participants with a Bachelors degree (Yoga, p = 0·041; Gita, p < 0·001; Yoga+Gita, p < 0·001), as well as for those with a Masters degree or higher (Yoga, p = 0·023; Gita, p = 0·049; Yoga+Gita, p = 0·018), the Friedman test showed significant improvements in GAD7 scores over time in all three groups. However, the control group remained largely unchanged in the participants with a Bachelors degree as well as those with a Masters degree or higher (p = 0·531 and p = 0·402, respectively).

**Table 4. Results of the Conover test.**

| Group | Comparison | Conover's Post-hoc p-values | Significant Difference |
|---|---|---|---|
| Yoga | B-GAD7 vs. IPI-GAD7 | 0·001 | Yes |
| Yoga | B-GAD7 vs. DPI-GAD7 | 0·001 | Yes |
| Yoga | IPI-GAD7 vs. DPI-GAD7 | 1·000 | No |
| Gita | B-GAD7 vs. IPI-GAD7 | 0·000 | Yes |
| Gita | B-GAD7 vs. DPI-GAD7 | 0·000 | Yes |
| Gita | IPI-GAD7 vs. DPI-GAD7 | 0·355 | No |
| Yoga+Gita | B-GAD7 vs. IPI-GAD7 | 0·000 | Yes |
| Yoga+Gita | B-GAD7 vs. DPI-GAD7 | 0·000 | Yes |
| Yoga+Gita | IPI-GAD7 vs. DPI-GAD7 | 1·000 | No |

*GAD7, generalised anxiety disorder-7; B-GAD7, baseline GAD7; IPI-GAD7, immediate post-intervention GAD7; DPI-GAD7, delayed post-intervention GAD7*

**Table 5. GAD7 values by gender and group over time.**

| Group | Gender | B-GAD7 (Median [IQR]) | Mann–Whitney U p-value | IPI-GAD7 (Median [IQR]) | Mann–Whitney U p-value | DPI-GAD7 (Median [IQR]) | Mann–Whitney U p-value | %Change (Baseline to 45 Days) (Median %Change [IQR]) | MCID Responders n | % |
|---|---|---|---|---|---|---|---|---|---|---|
| Yoga | Men | 6·000 [3·500, 8·500] | 0·838 | 5·000 [2·500, 7·500] | 0·959 | 4·000 [2·000, 6·500] | 0·839 | −30·000% [-50·000%, -8·333%] | 1 | 9·1% |
| Yoga | Women | 6·500 [6·000, 8·500] | ·· | 6·000 [5·000, 7·000] | ·· | 5·500 [2·750, 6·000] | ·· | −29·365% [-48·611%, -3·571%] | 2 | 33·3% |
| Gita | Men | 7·000 [4·500, 8·500] | 0·544 | 4·000 [3·000, 5·500] | 0·542 | 3·000 [2·000, 4·500] | 0·642 | −50·000% [-57·143%, -31·667%] | 4 | 36·4% |
| Gita | Women | 7·500 [6·250, 9·500] | ·· | 3·000 [2·000, 6·250] | ·· | 3·000 [2·250, 3·750] | ·· | −70·714% [-76·786%, -55·000%] | 4 | 66·7% |
| Yoga+Gita | Men | 6·500 [5·750, 7·750] | 0·383 | 5·000 [2·750, 5·500] | 0·067 | 4·000 [3·750, 5·000] | 0·242 | −41·667% [-57·857%, -19·167%] | 3 | 37·5% |
| Yoga+Gita | Women | 5·000 [4·000, 7·000] | ·· | 2·000 [2·000, 3·000] | ·· | 2·000 [1·000, 4·000] | ·· | −50·000% [-85·714%, -40·000%] | 4 | 44·4% |
| Control | Men | 8·000 [6·250, 8·750] | 0·169 | 7·000 [6·000, 8·000] | 0·489 | 6·500 [3·500, 7·750] | 0·493 | −14·583% [-38·333%, 0·000%] | 2 | 20·0% |
| Control | Women | 4·000 [3·000, 8·000] | ·· | 6·000 [3·000, 8·500] | ·· | 7·000 [4·000, 10·500] | ·· | 27·273% [-14·167%, 83·333%] | 0 | 0·0% |

*MCID, minimum clinically important difference (Responders are those with MCID >=4); GAD7, generalised anxiety disorder-7; B-GAD7, baseline GAD7; IPI-GAD7, immediate post-intervention GAD7; DPI-GAD7, delayed post-intervention GAD7; IQR, interquartile range.*

*MCID Responders are participants with ≥4-point reduction in GAD7 from Baseline to DPI (Day 52), denominator for calculating % = participants with both Baseline and DPI measurements. %Change is computed per participant as 100×(DPI−Baseline)/Baseline (participants with Baseline = 0 excluded), then summarized as median [IQR].*

**Table 6. GAD7 values by educational level and group over time.**

| Group | Education Level | B-GAD7 (Median [IQR]) | Mann–Whitney U p-value | IPI-GAD7 (Median [IQR]) | Mann–Whitney U p-value | DPI-GAD7 (Median [IQR]) | Mann–Whitney U p-value | %Change (Baseline to 45 Days) (Median %Change [IQR]) | MCID Responders | |
|---|---|---|---|---|---|---|---|---|---|---|
| | | | | | | | | | n | % |
| **Yoga** | Bachelors | 8·000 [6·000, 9·000] | 0·129 | 7·000 [5·000, 8·000] | 0·224 | 6·000 [4·000, 8·000] | 0·132 | −20·000% [-50·000%, 0·000%] | 1 | 11·1% |
| **Yoga** | Masters or higher | 6·000 [3·750, 6·000] | ·· | 4·500 [2·750, 5·500] | ·· | 2·500 [2·000, 5·000] | ·· | −38·889% [-50·000%, -12·500%] | 2 | 25·0% |
| **Gita** | Bachelors | 7·000 [4·500, 8·750] | 0·844 | 5·000 [3·000, 6·750] | 0·430 | 3·000 [2·000, 3·750] | 0·725 | −50·000% [-66·786%, -37·500%] | 4 | 40·0% |
| **Gita** | Masters or higher | 7·000 [5·500, 9·000] | ·· | 3·000 [2·500, 4·000] | ·· | 3·000 [2·500, 4·000] | ·· | −57·143% [-69·286%, -41·667%] | 4 | 57·1% |
| **Yoga+Gita** | Bachelors | 6·000 [5·000, 7·000] | 0·883 | 2·500 [2·000, 4·500] | 0·393 | 3·000 [1·250, 4·000] | 0·277 | −53·571% [-65·000%, -35·000%] | 4 | 40·0% |
| **Yoga+Gita** | Masters or higher | 5·000 [4·000, 8·500] | ·· | 3·000 [2·000, 6·000] | ·· | 5·000 [2·000, 5·500] | ·· | −46·667% [-75·000%, -17·143%] | 3 | 42·9% |
| **Control** | Bachelors | 4·500 [3·750, 5·750] | 0·037 | 6·000 [5·000, 6·250] | 0·263 | 4·500 [3·750, 6·500] | 0·192 | −10·000% [-35·000%, 27·083%] | 0 | 0·0% |
| **Control** | Masters or higher | 8·000 [7·000, 11·000] | ·· | 8·000 [4·000, 9·000] | ·· | 7·000 [7·000, 10·000] | ·· | −8·333% [-16·667%, 16·667%] | 2 | 22·2% |

*MCID, minimum clinically important difference (Responders are those with MCID >=4); GAD7, generalised anxiety disorder-7; B-GAD7, baseline GAD7; IPI-GAD7, immediate post-intervention GAD7; DPI-GAD7, delayed post-intervention GAD7; IQR, interquartile range.*

*MCID Responders are participants with ≥4-point reduction in GAD7 from Baseline to DPI (Day 52), denominator for calculating % = participants with both Baseline and DPI measurements. %Change is computed per participant as 100×(DPI−Baseline)/Baseline (participants with Baseline = 0 excluded), then summarized as median [IQR].*

**3.4.3. By continued practice.** In general, participants who continued practicing the interventions tended to show lower DPI-GAD7 scores than those who did not continue practicing the interventions (Table 7); however, statistical significance (p = 0·033) was observed only in the Yoga+Gita group.

## 4. Discussion

COVID-19 has posed serious challenges to the mental health of the HCWs [1,2,8]. Several interventions have been suggested to support the mental health of HCWs during the COVID-19 pandemic. Chiappetta et al. concluded that mindfulness-based interventions improved the mental wellbeing of the HCWs [19]. Another study showed that HCWs who had social support had better sleep quality, and prescribed employing psychotherapy teams to support the HCWs [20]. Yoga, relaxation, and meditation have also been recommended [4,15]. Some hospitals provided psychological support to HCWs through ensuring regular counsellor visits, training HCWs on how to relax, and encouraging leisure activities [21]. Hotlines for psychological assistance have been established to combat the mental health challenges of HCWs [22].

However, in a study of 300 HCWs with anxiety and depression, 59·8% participants said they would not consider seeking psychological support [8]. HCWs often feel ashamed of seeking help [23] and fear it may harm their reputation [9]. In India, mental health support systems for HCWs are largely lacking [24]. In our hospital, psychological support services were

**Table 7. Impact of continued practice on DPI-GAD7 scores.**

| Group | Practice Continued | N | DPI-GAD7 (Median [IQR]) | Mann–Whitney U p-value | %Change (Baseline to 45 Days) (Median %Change [IQR]) | MCID Responders | |
|---|---|---|---|---|---|---|---|
| | | | | | | n | % |
| Yoga | Yes | 12 | 4·500 [2·000, 6·250] | 0·873 | −37·222% [-51·389%, -16·071%] | 3 | 25·0% |
| Yoga | No | 5 | 6·000 [2·000, 6·000] | ·· | 0·000% [-33·333%, 0·000%] | 0 | 0·0% |
| Gita | Yes | 6 | 3·000 [3·000, 3·750] | 0·502 | −53·571% [-66·786%, -50·000%] | 4 | 66·7% |
| Gita | No | 11 | 3·000 [2·000, 4·000] | ·· | −50·000% [-65·714%, -33·333%] | 4 | 36·4% |
| Yoga+Gita | Yes | 11 | 3·000 [1·000, 4·000] | 0·033* | −60·000% [-87·302%, -45·000%] | 6 | 54·5% |
| Yoga+Gita | No | 6 | 5·000 [4·250, 5·750] | ·· | −18·333% [-40·000%, -14·881%] | 1 | 16·7% |

MCID, minimum clinically important difference (Responders are those with MCID >=4); GAD7, generalised anxiety disorder-7; DPI-GAD7, delayed post-intervention GAD7; IQR, interquartile range.

MCID Responders are participants with ≥4-point reduction in GAD7 from Baseline to DPI (Day 52), denominator for calculating % = participants with both Baseline and DPI measurements. %Change is computed per participant as 100×(DPI−Baseline)/Baseline (participants with Baseline = 0 excluded), then summarized as median [IQR].

underutilized despite significant stress during the pandemic. Evidently, HCWs have a negative attitude towards mental health issues and are reluctant to seek help. Therefore, an intervention that does not appear like a psychological intervention at the outset can act as a workaround solution. Given prior findings that HCWs turned to spirituality as a coping mechanism during the COVID-19 pandemic [2], we sought to provide a spirituality-based intervention to support their mental health.

In this study, there were no significant differences in age, educational attainment, gender, and B-GAD7 scores, among the groups at baseline, suggesting that any observed effects of the interventions are less likely to be confounded by these baseline characteristics. The results showed significant reductions in GAD7 scores from baseline to immediately post-intervention and from baseline to 45 days post-intervention in all intervention groups, underscoring the effectiveness of Yoga and the Bhagavad Gita teachings, alone and in combination, in mitigating stress. Contrarily, the lack of changes in GAD7 scores over time in the control group reinforce that the improvements in the GAD7 scores in the intervention groups are directly attributable to the interventions. Moreover, the sustained reductions in GAD7 scores 45 days post-intervention, particularly in the Gita and Yoga+Gita groups, underscore the lasting impact of these interventions on reducing stress. The large effect sizes for IPI-GAD7 and DPI-GAD7 indicate that the Yoga+Gita and Gita-alone interventions have a substantial practically significant impact on reducing stress among HCWs compared to the control group, highlighting the potential for integrating these traditional approaches into therapeutic contexts.

Interestingly, only the Yoga+Gita group showed a significant reduction in IPI-GAD7 scores and only the Gita group showed a significant reduction in DPI-GAD7 scores when the GAD7 scores at each timepoint were compared across the groups. This result underscores the complementary nature of Yoga and Gita. While Yoga offers immediate physical relief through breath control and relaxation, this alone may not be sufficient to produce significant stress reduction in a short span of 1 week. Indeed, most Yoga-alone interventions for stress reduction span a longer period of 4 weeks [25]. The Bhagavad Gita provides deeper emotional and cognitive benefits that foster long-term resilience. The combination of Yoga and Gita leads to both immediate and sustained stress reduction, as Yoga addresses body discomfort and tension by shifting the balance from the sympathetic to parasympathetic nervous system [26], and the Gita's teachings promote a shift in mindset

and emotional stability. The Gita alone works more gradually, explaining its impact after 45 days, as spiritual and cognitive frameworks take time to reshape attitudes and reduce distress. Hence, there is a need to integrate physical and spiritual practices for comprehensive mental health benefits. The Kendall's W values indicate that the interventions lead to moderate–strong reductions in psychological distress over time within each intervention group, validating their effectiveness.

In exploratory responder analyses, the odds of achieving a clinically meaningful reduction in GAD7 (≥4 points) were higher in the Gita arm compared with Control (OR = 6.67, 95% CI 1.15–38.60), with a similarly elevated but imprecise estimate observed in the Yoga+Gita arm (OR = 5.25, 95% CI 0.90–30.62). In contrast, Yoga alone showed little difference relative to Control (OR = 1.61, 95% CI 0.23–11.09). These effect size estimates suggest that inclusion of structured Bhagavad Gita-based cognitive reframing may increase the likelihood of clinically meaningful anxiety reduction among healthcare workers. However, confidence intervals were wide, reflecting limited precision due to the small sample size. The study was not powered for categorical responder outcomes; therefore, these findings should be interpreted as exploratory and hypothesis-generating rather than definitive evidence of superiority. Larger trials powered specifically for categorical responder outcomes are required to confirm these estimates.

Women demonstrated a more substantial reduction in GAD7 scores, particularly in the Gita and Yoga+Gita groups, suggesting that while both men and women benefit from these interventions, women might experience more significant improvements. This result is similar to that of a previous study which revealed that women showed a greater reduction in negative affect than men in response to meditation [27]. Therefore, considering gender as a factor in the design and evaluation of religious/spiritual interventions (RSIs) may be important. In terms of educational attainment, there were no differences in the responses of any group to the interventions. Thus, educational attainment may not influence the effectiveness of Yoga and Gita, alone and in combination, in mitigating stress. In general, participants who continued practicing the interventions tended to show lower DPI-GAD7 scores than those who did not; however, significance was reached only in the Yoga+Gita group, suggesting a potentially synergistic effect of continuing both practices.

RSIs can improve mental wellbeing by giving people a sense of purpose and acting as a coping mechanism to mitigate stress. Hook et al. [28] have opined that for psychological problems, RSIs may be more effective than secular therapies and even some drug treatments. Several studies on RSIs for mental health conditions have demonstrated a benefit of RSIs, especially at 1–3 months of follow-up [29,30]. Similar to these studies, our findings also show a significant benefit of Yoga and Gita, alone and in combination, on the mental wellbeing of HCWs, both immediately after the intervention and after 45 days. However, these previous studies on RSIs involved only Christian, Islamic, or Jewish practices and participants. Our study, on the contrary, examines Hindu RSIs (practicing Yoga and learning the Bhagavad Gita) and all our participants, except one, were practicing Hindus. Another main distinction of our study is that our RSI was relatively short, lasting 1 week, but showed significant sustained beneficial effects. Furthermore, to the best of our knowledge, this is the first study to explore the effectiveness of RSIs to support the mental health of HCWs.

The limitations of the study include a relatively smaller sample size due to challenges in recruiting HCWs at a time when they were already working in overstretched shifts. Because the sample size was calculated in PASS using the default two-group comparison, the trial may be underpowered to detect all pairwise differences across the three intervention arms and the control arm after multiplicity correction. Accordingly, some pairwise comparisons—particularly among intervention arms—may be subject to type II error. Subgroup and continued-practice analyses were exploratory and unadjusted for multiple comparisons, and therefore carry an increased risk of type I error. Consequently, the findings have limited external generalizability beyond similar single-center, secondary-care hospital settings and comparable HCW populations. Although linear mixed models or generalized estimating equations would model longitudinal correlation more efficiently, given the small sample size and the multi-arm repeated-measures design, mixed-model or generalized estimating equation estimates—particularly for group-by-time effects—would be imprecise and potentially unstable; we therefore used nonparametric tests given the observed non-normality. To address this limitation, our ongoing larger trial of these interventions in a different target population is powered specifically for mixed-model analysis [31].

Because of the nature of the interventions, participants were aware of their group assignment, which may have introduced expectancy or performance bias, particularly in the spiritually based intervention arms. However, outcome assessors and data analysts were blinded to group allocation, and participants were informed that all interventions were expected to reduce stress, which may have mitigated differential expectancy effects across groups. Adherence ('continued practice') was measured only with a single self-reported yes/no question and did not capture how often, how long, or how accurately participants practiced; this coarse measure may misclassify adherence and limits causal interpretation of sustained effects. Future studies should incorporate objective or structured adherence tracking (e.g., practice logs, digital monitoring, or instructor verification). In addition, use of a usual-care control does not account for nonspecific attention effects; future studies may benefit from including an attention-matched comparator arm.

Additionally, the sample is not broadly representative as it largely consists of participants belonging to the Hindu Religion, thereby limiting the applicability of the evaluated interventions. Analogous spiritual texts or secular contemplative practices in non-Hindu contexts may offer similar cognitive-emotional benefits and warrant systematic evaluation in future trials. Furthermore, the study may be affected by recruitment bias as HCWs who chose to participate in the study may be inherently spiritual/religious. Moreover, the inherent religiosity/spirituality of the participants may have confounded the results, and because we could not measure the baseline inherent religiosity/spirituality of the participants, we could not account for this potential confounder in our analyses. It remains to be seen how these interventions affect the HCWs belonging to other faiths and those who are agnostic to religion and spirituality, before generalizing these findings to all HCWs. We aim to flesh out the impact of the baseline spiritual/religious beliefs of the participants on the effectiveness of RSIs in a future study.

## 5. Conclusion

Psychological support systems have gone largely underutilized by HCWs, necessitating novel approaches to supporting their mental wellbeing. The findings herein underscore the significant positive impact of learning the Bhagavad Gita, alone and in combination with practicing Yoga, on the mental wellbeing of HCWs, suggesting that these approaches may offer effective strategies for managing psychological distress.

## Supporting information

**S1 File. CONSORT 2010 checklist of information to include when reporting a randomised trial.**
(DOCX)

**S2 File. Study protocol v4 (13.09.2021).**
(DOCX)

## Author contributions

**Conceptualization:** Laalithya Konduru.

**Data curation:** Nishant Das, Simranjeet Singh Dahia.

**Formal analysis:** Laalithya Konduru.

**Funding acquisition:** Nishant Das, Gargi Kothari-Speakman.

**Investigation:** Laalithya Konduru, Nishant Das, Gargi Kothari-Speakman.

**Methodology:** Laalithya Konduru, Simranjeet Singh Dahia, Santwana Sagnika.

**Project administration:** Nishant Das, Ajit Kumar Behura.

**Resources:** Nishant Das, Gargi Kothari-Speakman, Ajit Kumar Behura.

**Software:** Simranjeet Singh Dahia, Santwana Sagnika.

**Supervision:** Laalithya Konduru.

**Validation:** Laalithya Konduru, Santwana Sagnika.

**Visualization:** Simranjeet Singh Dahia.

**Writing – original draft:** Laalithya Konduru, Nishant Das.

**Writing – review & editing:** Simranjeet Singh Dahia, Santwana Sagnika, Gargi Kothari-Speakman, Ajit Kumar Behura.

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
