## [Decision Letter · Decision Letter 0]

23 Jun 2025

Dear Dr. Konduru,

Thank you for submitting your manuscript to PLOS ONE. After careful consideration, we feel that it has merit but does not fully meet PLOS ONE’s publication criteria as it currently stands. Therefore, we invite you to submit a revised version of the manuscript that addresses the points raised during the review process.

We look forward to receiving your revised manuscript.

Kind regards,

Satish G Patil, PhD

Academic Editor

PLOS ONE

Journal Requirements:

2. Registration done retrospectively (after enrollment of participants) (TC2/PRTC Note)

Thank you for submitting your clinical trial to PLOS ONE and for providing the name of the registry and the registration number. The information in the registry entry suggests that your trial was registered after patient recruitment began. PLOS ONE strongly encourages authors to register all trials before recruiting the first participant in a study.

1) your reasons for your delay in registering this study (after enrolment of participants started);

2) confirmation that all related trials are registered by stating: “The authors confirm that all ongoing and related trials for this drug/intervention are registered”.

Reviewers' comments:

Reviewer's Responses to Questions

**Comments to the Author**

1. Is the manuscript technically sound, and do the data support the conclusions?

Reviewer #1: Partly

Reviewer #2: Yes

Reviewer #3: Yes

2. Has the statistical analysis been performed appropriately and rigorously?

Reviewer #1: No

Reviewer #2: Yes

Reviewer #3: Yes

3. Have the authors made all data underlying the findings in their manuscript fully available?

Reviewer #1: Yes

Reviewer #2: Yes

Reviewer #3: Yes

4. Is the manuscript presented in an intelligible fashion and written in standard English?

Reviewer #1: Yes

Reviewer #2: Yes

Reviewer #3: Yes

Reviewer #1: Was a t-test used for the sample size calculation?

Consider performing an analysis of the proportion of participants who achieved the minimum clinically important difference of 4 by treatment group.

If a nonparametric method was used, it would be better to report the median and IQR instead.

To analyze the longitudinal data, a linear mixed model or GEE approach is recommended. These methods provide more statistical power and are more robust to violations of normality assumptions. Within-group and between-group analyses can be included in one model.

Hypothesis testing is not recommended for subgroup analyses, as these analyses are underpowered for such a small sample size. Non significant results are inconclusive. Even statistically significant findings may be due to chance, and appropriate adjustments for multiple comparisons would be necessary.

A figure displaying group means over time is recommended.

Reviewer #2: 2. Major Comments

2.1. Study Design & Randomization

1. Randomization Method (Chit-Pull)

o Issue: Simple randomization in small samples (n = 68) can yield baseline imbalances by chance. Although the authors report no significant differences in age, gender, or education (ANOVA p > 0.7 for all), they did not present the baseline GAD-7 distributions across groups in order to confirm statistical equivalence on the primary outcome variable.

2. Sample Size Calculation & Power

o The authors used PASS 2020 to calculate n = 68 (17 per arm) based on: α = 0.05, power = 80%, minimum clinically important difference for GAD-7 of 4, SD = 3.78 (from a previous study), and a 15% dropout.

o Clarification Needed: It would be helpful to specify whether the sample size calculation was powered to detect differences between each pair of groups or only to detect a difference between “any intervention” versus control. Given four arms, a calculation designed for a two-group comparison may not yield adequate power for pairwise comparisons among three intervention arms.

o Recommendation: Briefly clarify in Methods whether the calculation aimed for (a) a two-group comparison (e.g., Gita vs. control) or (b) multiple comparisons (e.g., adjusting for 3 pairwise tests). If the latter, describe any adjustment (such as Bonferroni) made or comment on the potential for type II error in some comparisons.

3. Masking / Blinding

o The trial was open-label for participants (because interventions are behavioral), but the data analysts were blinded. This is acceptable.

o Recommendation: In the Discussion, acknowledge that open-label assignment may introduce performance bias (e.g., expectation effects), especially in spiritually oriented interventions. If possible, note any steps taken to minimize this (for instance, emphasizing to participants that “all interventions are hypothesized to reduce stress,” whatever the arm).

2.2. Intervention Details & Adherence

1. Yoga Protocol

o The manuscript lists specific asanas and pranayama (e.g., Bhadrasana, Shawasana, Nadi Shodhana Pranayama, etc.), delivered by certified teachers.

o Minor Point: The description could specify whether each session included a standardized sequence (e.g., warm-up, standing postures, seated postures, pranayama, meditation) and how long each component lasted.

o Recommendation: Add a brief sentence (Methods §2.2) confirming that each one-hour session followed a standardized format (e.g., “Each session began with a 5-minute centering meditation, 20 minutes of asana practice, 20 minutes of pranayama, and concluded with 15 minutes of guided meditation”). This will help reproducibility.

2. Bhagavad Gita Protocol

o Participants in Gita and Yoga+Gita attended one-hour daily sessions learning and discussing Chapter 2 of the Gita.

o Minor Point: It would be useful to know who delivered these sessions (e.g., trained teachers of ISKCON), how they structured learning (reading aloud, discussion, Q&A), and whether participants received any written materials or audio recordings to reinforce learning during follow-up.

o Recommendation: Clarify in Methods if handouts/translations were provided, whether the same instructor led each session, and if participants were encouraged to review the verses during the 45-day follow-up period (beyond simply asking whether they “continued practice”).

3. Measurement of Continued Practice

o The authors divided intervention groups into subgroups (“continued practice” vs. “did not continue”) at 45 days. However, Table 7 (p. 28) shows that in the Yoga+Gita group, those who did not continue practice had a lower DPI-GAD7 (2.455 ± 1.753) than those who did continue (4.833 ± 2.317), yet the text (Results §3.4.3) states that continued practice was associated with lower DPI-GAD7 scores.

o Major Concern: This apparent discrepancy between text and table must be resolved. It is unclear whether the table cells have been erroneously swapped, whether the text wrongly interprets the table, or whether a coding error occurred.

o Recommendation: Please double-check the raw data and clarify which subgroup’s mean is which. If indeed those who continued had lower stress scores, the table’s labels need to be reversed; if not, correct the text to reflect the data. In either case, add a brief explanation (e.g., “Participants who reported continuing the practices for 45 days scored X vs. Y in DPI-GAD7 (p = .032), indicating a sustained benefit among adherents”).

2.3. Statistical Analysis

1. Choice of Nonparametric Tests

o The Shapiro–Wilk test indicated that demographic variables were normally distributed but GAD-7 scores were not, justifying Friedman (within-group) and Kruskal–Wallis (between-group) tests. Pairwise comparisons used Conover (for Friedman) and Dunn’s (for Kruskal–Wallis).

o Strength: This is appropriate for small samples with non-normal distributions. The use of effect sizes (η² for between-groups; Kendall’s W for within-groups) is commendable.

o Issue: The manuscript does not explicitly state whether p-values were adjusted for multiple pairwise comparisons (for instance, if Dunn’s test used a Bonferroni correction or a false discovery rate). Although many software packages automatically adjust, this should be explicitly noted.

o Recommendation: In the Statistical Analysis subsection, specify whether and how Dunn’s and Conover’s tests were adjusted for multiple comparisons. If no adjustment was made, comment on the possibility of inflated type I error rates and whether findings near the 0.05 threshold (e.g., p = 0.047 for Gita vs. control at DPI) remain robust after correction.

2. Presentation of Effect Sizes

o The authors report η² = 0.166 for IPI-GAD7 and η² = 0.140 for DPI-GAD7.

o Minor Point: It may be helpful to remind readers of conventional benchmarks (e.g., small = 0.01, medium = 0.06, large = 0.14 for η²; or describe Kendall’s W magnitude).

o Recommendation: Briefly state in the Results (e.g., “η² = 0.166 indicates a large effect size according to Cohen’s conventions for nonparametric tests”) to assist interpretation.

2.4. Results & Data Presentation

1. Baseline Comparisons

o Although Table 1 shows no significant baseline differences in demographics, baseline GAD-7 by group is not explicitly compared.

o Recommendation (echoing §2.2.1): Add a row in Table 2 or a supplementary table indicating baseline (B-GAD7) differences (median and IQR) with a corresponding Kruskal–Wallis p-value. If not significant, a brief statement in the text reiterating “Baseline GAD-7 did not differ significantly across groups (Kruskal–Wallis p = 0.908)” would suffice.

2. Tables & Figures

o Table 2 succinctly presents mean ± SD and percent change; however, percent change uses the formula “100×((DPI-GAD7)−(B-GAD7))/(B-GAD7)”. In the equation printed under the table, there is a missing “7” in the denominator (i.e., “(B-GAD−7)” rather than “(B-GAD7)”).

o Recommendation: Correct the typographical error in the formula and ensure consistency (e.g., “(B-GAD7)” without stray hyphens).

o Table 7 Labeling Discrepancy (discussed in §2.2.3). Once clarified, ensure that subheadings (Yes vs. No) clearly correspond to “Continued practice” versus “Did not continue.”

o Figure 1 (CONSORT Flowchart) is present and correctly depicts participant flow. Ensure the figure caption briefly states total assessed (n = 79), excluded (n = 11), randomized (n = 68), and final analysis (n = 68), per CONSORT guidelines.

3. Subgroup Analyses

o The authors present subgroup analyses by gender (Table 5) and education (Table 6). Although interesting, these stratified tests inflate type I error risk.

o Recommendation: Acknowledge in the Discussion (Limitations) that subgroup findings (especially gender differences) are exploratory and warrant replication in larger, adequately powered samples. Emphasize that these results should be interpreted with caution.

2.5. Discussion & Interpretation

1. Integration with Prior Literature

o The Discussion situates findings within existing research on RSIs (religious/spiritual interventions) and mindfulness for HCWs (e.g., Chiappetta et al., Hook et al.), which is appropriate.

o Strength: The authors clearly articulate how yoga addresses immediate physiological stress (parasympathetic activation) while the Gita fosters cognitive‐emotional shifts, supporting the complementary hypothesis.

2. Generality & Cultural Context

o The manuscript acknowledges that participants were almost all practicing Hindus, limiting generalizability to HCWs of other faiths or secular orientations.

o Recommendation: Consider adding a sentence on potential cultural transferability—e.g., whether equivalent spiritual texts or meditation practices from other traditions might yield similar benefits. This will help readers in non-Hindu contexts appreciate how the model might be adapted.

3. Limitations

o The authors list sample size constraints, recruitment bias (self-selection by spiritually inclined HCWs), and lack of religiosity measurement at baseline.

o Additional Limitation: Since adherence beyond self-report (“continued practice”) was binary and based on participant recall, there is potential misclassification. Suggest acknowledging that objective measures (e.g., practice logs or digital tracking) would strengthen future trials.

o Recommendation: In Limitations, add:

“Because ‘continued practice’ was assessed via self-report at 45 days, social desirability or recall bias may have affected subgroup classification. Future RCTs could incorporate practice diaries or digital reminders to more objectively assess long-term adherence.”

2.6. Ethical & Reporting Considerations

1. Ethics Statement

o Ethics approval (SJHRC–NI/21/OCT/07) and written informed consent are clearly stated (Methods §2.2 and Ethics Statement).

o Minor Point: Confirm that the Ethics Statement in submission metadata exactly mirrors the Methods section (including the consent type and any waiver if applicable).

o Recommendation: In the submission form, ensure “Written informed consent was obtained” appears under “Ethics Statement” verbatim.

2. Data Availability

o The authors state that “All relevant data are within the manuscript and its Supporting Information files,” with a Figshare DOI (https://doi.org/10.6084/m9.figshare.27285234) for individual participant data and dictionary, plus OSF registration info. This aligns with PLOS ONE policy.

o Recommendation: Consider specifying if the Figshare repository is publicly accessible immediately upon publication or requires any embargo/permissions. If there is no restriction, a statement such as “The Figshare dataset (DOI: …) is publicly accessible without restriction” would be ideal.

3. Reporting Guidelines

o The inclusion of a completed CONSORT flowchart (Fig 1) is appropriate. However, the manuscript would benefit from an explicit statement that the reporting adheres to the CONSORT 2010 checklist.

o Recommendation: In Methods §2 (or at the end of the Introduction), add:

“Reporting of this RCT follows the CONSORT 2010 guidelines (Schulz, Altman, & Moher, 2010). A completed CONSORT checklist is provided as Supporting Information.”

Reviewer #3: Strengths of this Manuscript:

• The study addresses a significant gap by evaluating culturally tailored interventions for HCWs in India, where Western mental health strategies may not be fully applicable.

• The use of a randomized controlled trial design with a control group strengthens the validity of the findings.

• The interventions are well-described, and the use of validated tools (GAD-7) for outcome measurement is appropriate.

• The manuscript highlights the importance of cultural sensitivity in mental health interventions, which is a valuable contribution to the literature.

Major Comments

1. Sample Size and Generalizability: The study included 68 participants (17 per group), which is relatively small. While a power calculation was performed, the limited sample size and single-center design may restrict the generalizability of the findings. Please discuss this limitation more explicitly in the discussion section.

2. Randomization and Blinding: The chit-pull method for randomization is described, but more details on allocation concealment and how potential selection bias was minimized would strengthen the methodology section.

Participants were aware of their group assignments, which could introduce performance bias. While data analysts were blinded, the lack of participant blinding should be acknowledged as a limitation.

3. Statistical Analysis

The manuscript mentions the use of non-parametric tests due to non-normal GAD-7 distributions. Please clarify whether adjustments for multiple comparisons were made during subgroup analyses to control for type I error.

It would be helpful to report effect sizes alongside p-values to provide a sense of the practical significance of the findings.

4. Intervention Fidelity and Adherence

The study mentions that participants were asked about continued practice at follow-up. Please provide more detail on how adherence to the interventions was monitored and whether this influenced the outcomes.

5. Control Group Considerations

The control group received no intervention. Consider discussing whether an attention control or placebo intervention could have helped account for the effects of group participation and attention from facilitators.

Minor Comments

• The abstract and introduction could benefit from more precise language regarding the hypotheses and objectives.

• Please clarify the rationale for selecting only the second chapter of the Bhagavad Gita for the intervention.

• Some references in the introduction are cited as numbers (e.g., , ) but the reference list is not included. Ensure all references are properly listed and formatted.

• There are minor typographical and grammatical errors throughout the manuscript that should be addressed during proofreading.

The manuscript presents promising findings on the use of culturally tailored interventions for reducing psychological distress among HCWs during the COVID-19 pandemic. Addressing the points above will strengthen the manuscript and its contribution to the field.

.

Reviewer #1: No

Reviewer #2: No

Reviewer #3: **Yes:** Dr D Elanchezhiyan, Research Officer, CCRYN, New DelhiDr D Elanchezhiyan, Research Officer, CCRYN, New DelhiDr D Elanchezhiyan, Research Officer, CCRYN, New DelhiDr D Elanchezhiyan, Research Officer, CCRYN, New Delhi

---

## [Author Response · Author response to Decision Letter 1]

1 Mar 2026

We have made the necessary changes. A point-by-point response file has been uploaded for more details.

---

## [Decision Letter · Decision Letter 1]

1 Apr 2026

The Stress Reduction Potential of Bhagavad Gita and Yoga for Healthcare Workers During The COVID-19 Pandemic: A Randomized Controlled Trial

PONE-D-25-12535R1

Dear Dr. Konduru,

We’re pleased to inform you that your manuscript has been judged scientifically suitable for publication and will be formally accepted for publication once it meets all outstanding technical requirements.

Kind regards,

Satish G Patil, PhD

Academic Editor

PLOS One

Reviewers' comments:

Reviewer's Responses to Questions

**Comments to the Author**

Reviewer #1: All comments have been addressed

Reviewer #2: All comments have been addressed

Reviewer #3: All comments have been addressed

2. Is the manuscript technically sound, and do the data support the conclusions?

Reviewer #1: (No Response)

Reviewer #2: Yes

Reviewer #3: Yes

3. Has the statistical analysis been performed appropriately and rigorously?

Reviewer #1: (No Response)

Reviewer #2: Yes

Reviewer #3: Yes

4. Have the authors made all data underlying the findings in their manuscript fully available?

Reviewer #1: (No Response)

Reviewer #2: Yes

Reviewer #3: Yes

5. Is the manuscript presented in an intelligible fashion and written in standard English?

Reviewer #1: (No Response)

Reviewer #2: Yes

Reviewer #3: Yes

Reviewer #1: All concerns are addressed.

Reviewer #2: The Author has addressed all my Suggestions. However it may be accepted in present from if any concern not raised by the Editor as well as another author.

Reviewer #3: The author has carefully revised the manuscript incorporating all comments and suggestions provided in the previous review round.

.

Reviewer #1: No

Reviewer #2: No

Reviewer #3: **Yes:** Dr D Elanchezhiyan, Research Officer, CCRYN, New DelhiDr D Elanchezhiyan, Research Officer, CCRYN, New DelhiDr D Elanchezhiyan, Research Officer, CCRYN, New DelhiDr D Elanchezhiyan, Research Officer, CCRYN, New Delhi

---

## [Editor Report · Acceptance letter]

PONE-D-25-12535R1

PLOS One

Dear Dr. Konduru,

I'm pleased to inform you that your manuscript has been deemed suitable for publication in PLOS One. Congratulations! Your manuscript is now being handed over to our production team.

Kind regards,

on behalf of

Prof. Dr. Satish G Patil

Academic Editor

PLOS One